# Rock Mass Characterization by UAV and Close-Range Photogrammetry: A Multiscale Approach Applied along the Vallone dell'Elva Road (Italy)

**Maria Migliazza [1], Maria Teresa Carriero [1,\*], Andrea Lingua [2] , Emanuele Pontoglio [2] and Claudio Scavia [1]**

[1] Department of Structural, Geotechnical and Building Engineering, Politecnico di Torino, Corso Duca degli Abruzzi 24, 10129 Torino, Italy; maria.migliazza@polito.it (M.M.); claudio.scavia@polito.it (C.S.)

[2] Department of Environment, Land and Infrastructure Engineering, Politecnico di Torino, Corso Duca degli Abruzzi 24, 10129 Torino, Italy; andrea.lingua@polito.it (A.L.); emanuele.pontoglio@polito.it (E.P.)

**\*** Correspondence: maria.carriero@polito.it

**Abstract:** Geostructural rock mass surveys and the collection of data related to discontinues provide the basis for the characterization of rock masses and the study of their stability conditions. This paper describes a multiscale approach that was carried out using both non-contact techniques and traditional support techniques to survey certain geometrical features of discontinuities, such as their orientation, spacing, and useful persistence. This information is useful in identifying the possible kinematics and stability conditions. These techniques are extremely useful in the case study of the Elva valley road (Northern Italy), in which instability phenomena are spread across 9 km in an overhanging rocky mass. A multiscale approach was applied, obtaining digital surface models (DSMs) at three different scales: large-scale DSM of the entire road, a medium-scale DSM to assess portions of the slope, and a small-scale DSM to assess single discontinuities. The georeferenced point cloud and consequent DSMs of the slopes were obtained using an unmanned aerial vehicle (UAV) and terrestrial photogrammetric technique, allowing topographic and rapid traditional geostructural surveys. This technique allowed us to take measurements along the entire road, obtaining geometrical data for the discontinuities that are statistically representative of the rock mass and useful in defining the possible kinematic mechanisms and volumes of potentially detachable blocks. The main purpose of this study was to analyse how the geostructural features of a rock mass can affect the stability slope conditions at different scales in order to identify road sectors susceptible to different potential failure mechanisms using only kinematic analysis.

**Keywords:** slope stability; geostructural survey; point cloud; photogrammetry; UAV technique

## 1. Introduction

Slope stability conditions are strongly affected by discontinuities in geometrical features, which control the possible kinetic conditions of the potentially detachable blocks, their volumes, and their stability conditions. In addition to the orientation of discontinuities influencing the kinematic mechanism that can occur (planar, three-dimensional sliding, rockfall, toppling, etc.), their spacing and persistence greatly affect the shape and volume of potentially unstable blocks [1,2]. The surface morphology (roughness) of a discontinuity controls the shear strength and subsequent stability conditions. Furthermore, the potential kinematic mechanism is conditioned by the orientation of the rock face, which can vary along a slope and affect the local stability conditions of the rock mass.

Any engineering activities aimed at defining rock mass stability conditions or the extent of the risk associated with them must include extensive characterization of the rock masses, with a systematic and representative survey of the discontinuities, as well

as a series of investigations and tests to evaluate their mechanical characteristics. The characterization of rock masses is affected by the definition of the representative elementary volume (REV), which indicates the minimum dimensions so that the defined properties (either mechanical or geometric) are representative of the analysed medium [3,4]. Often, the rock mass discontinuity network plays a fundamental role in the stability of a rock slope, meaning the measurements carried out during a geostructural survey must be statistically representative of the entire rock mass.

As such, the survey techniques can be divided into two macrocategories: traditional techniques and indirect techniques. Usually, geostructural evaluation surveys are carried out using traditional techniques, in which the operator, by means of a geological compass and metric strip, is in direct contact with the rock face. Through the traditional survey approach, it is possible to obtain direct measurements of the orientations of discontinuities and other data relating to the discontinuities, such as filling, degree of alteration, aperture, roughness, and wall resistance data. Despite the traditional geostructural survey approach being the most widely used and well-known method, in some cases, it may present certain limitations, due to the fact that the measurements are localized, resulting in errors due to human and instrumental inaccuracies, while environmental conditions can also influence their representativeness [5].

The indirect or non-contact survey approaches are methods in which the measurements are not directly made on the rock wall but are carried out remotely using photogrammetric and light detection and ranging (LiDAR) techniques. They allow the morphology of the slopes to be measured through the creation of point clouds belonging to the exposed surface of the rock masses [6–14]. These techniques have been used since the end of the last century and have further increased in popularity with the appearance on the market of UAVs, which have expanded and facilitated the use of these techniques, even in complex situations [15–20]. These approaches allow non-direct measurements of the geometrical characteristics of the discontinuities derived from the analysis of an acquired three-dimensional (3D) point cloud of the terrain surface, which can be obtained with high accuracy in a short amount of time and over a wide area. This progress has also changed the way larger amounts of data regarding the geostructural conditions of large areas are collected and interpreted [4–8], as well as the consequent stability condition assessments [21–27], with the great advantage of allowing for inaccessible steep slopes to be surveyed in safe conditions. To become a real alternative (both in terms of productivity and accuracy) to the traditional survey approach, these techniques require interactive or automated software tools, which would allow the efficient selection in the point cloud of elements of interest (i.e., discontinuity planes and traces).

This study describes a multiscale approach to the study of the possible conditions of detachment for the Elva valley road (Northern Italy), in which both non-contact techniques and traditional support techniques are applied. The images acquired using UAVs and the terrestrial photogrammetric technique are used to obtain 3D point clouds, and high-resolution georeferenced DSMs are analysed to extract orientation and persistence data for the discontinuities at different scales. As described below, a multiscale approach was applied, obtaining DSMs at three different scales: a large-scale DSM of the entire road was produced, medium-scale models of the slope portions were obtained to identify the orientation of the discontinuities, and a small-scale DSM (equal to the scale of a single discontinuity) was used to obtain higher resolution point clouds in order to measure the roughness of the discontinuities. This paper describes the surveys and the results obtained at large and medium scales and focuses on the definition of orientation, spacing and persistence data, which is useful in defining the possible kinematic conditions and for volume estimates of the potentially detachable blocks. This article does not involve an analysis of the obtained roughness data.

For the case study, the surveys were carried out, the results of which are described below.

## 2. The Vallone dell'Elva Road

The so-called "Strada del Vallone" is located in the orographic left of the Maira Valley (Piedmont, Northern Italy) and directly connects the village of Elva (located at about 1637 m a.s.l.) with the Maira valley bottom road.

The road stretches for about 9 km of carriageway with overhanging rock walls and twelve tunnels excavated into the bare rock (Figure 1). Due to the numerous instability phenomena that have occurred since its construction (Table 1) along the entire road and after a large planar sliding in 2015, the road has been closed to traffic.

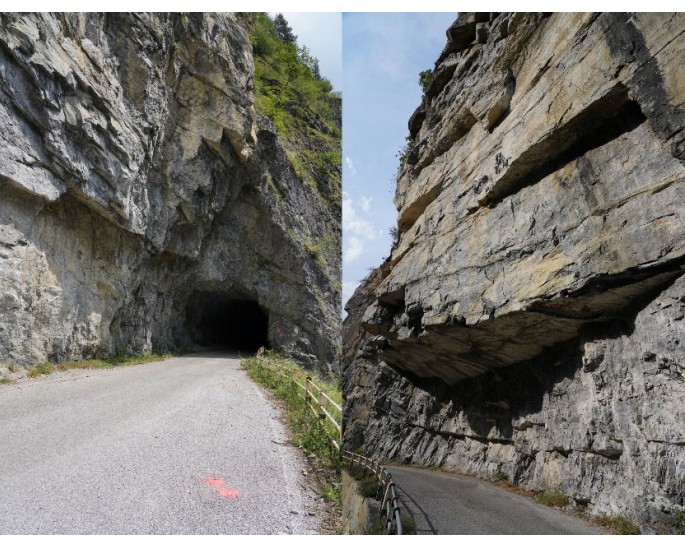

**Figure 1.** Photographs of the rock walls under study.

The Elva Road was built in 1883 as an impervious path, of one meter wide, running through the rock walls and along the gullies, dug with the use of explosives [28]. In 1934, the road saw its first widening in order to allow the transit of products and goods, and in 1956, its geometry was substantially modified, widening the roadway, modifying the route and creating the current tunnels and the actual road configuration [29], in order to make it passable by the vehicles.

The Elva Road is located in an area strongly tectonized with hectometric folds clearly visible in the panorama of Mount Bittone and delimited by a fault oriented toward N–S and passing through the right side of the valley itself (Figure 2). It is characterized (Figure 3 geological map extract) by dolomite rocks and light grey or black dolomitic limestones, in banks of considerable thickness, and interlayers of cinerites. Their formation took place in the Carnic and Noric Alps (Upper Triassic). These limestone–dolomitic carbonate rocks, cut from the road, are highly verticalized, with a high-angle westward immersion. This sedimentary rock is characterized by three systems, a highly persistent bedding plane and at least two more conjugated set joints, contained between, and perpendicular to, two continuous bedding planes. The orientation of the planes varies due to the presence of folds and faults, and for this reason different kinematic conditions are identified to also be linked to the variation of the orientation of the slopes along the road. The summit portion of the Valley near the town of Elva is instead set in the calcschists and micascists, with significant areas covered with soil that are the site of a deep gravitational slope deformation.

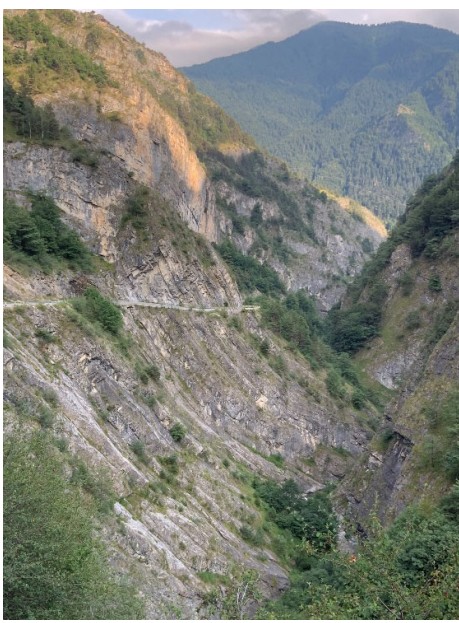

**Figure 2.** View of the hetometric fold: photo taken along the road looking south, and the road is visible on the right side of the slope.

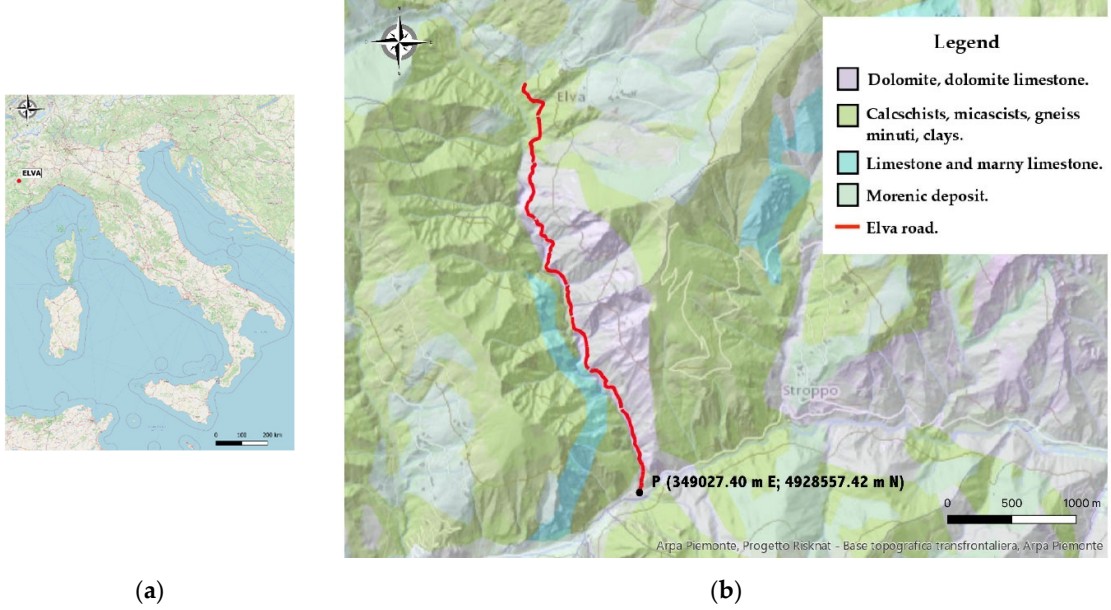

(**a**)                                                                                           (**b**)

**Figure 3.** (**a**) Map of Italy with indication of the position of Elva (coordinates of ELVA UTM zone 32T 347,654.28 m E; 4,934,405.34 m N); (**b**) geological map extract of the Vallone dell'Elva [30]; coordinates of point P (UTM zone 32T, 349,027.40 m E, 4,938,557.42 m N).

The Elva Valley Road is characterized by an instability that spans 9 km, with an overhanging rocky mass and the possibility of the detachment of blocks and kinematics that vary along the whole road. From the data reported in Table 1, it is clear that, since its construction, the road has always highlighted phenomena of natural instability occurrences or those induced by its construction [31]: some involved the detachment of individual blocks of variable volume; others involved larger portions of rock mass. If in the provincial archives, there is information on the phenomena of greater intensity, on the other hand, the citizens, who drive along the road daily, report indications of the detachment of single blocks that with fall on the roadway monthly [31,32].

From the information collected, it is possible to distinguish widespread instability phenomena involving single blocks with volume variables between a few cm$^3$ to about 1 m$^3$ occurring along the entire road with a high temporal frequency. In addition, localized instability phenomena, with higher intensity in terms of unstable rock mass volume (variable between a few m$^3$ to 2500 m$^3$) was recorded by the public agency (Table 1), which led, in some cases, to the interruption of the traffic along the road (Figure 4).

Depending on the discontinuity sets and slope orientation that vary along the entire slope, the instability phenomena that characterize the rock mass are of various kinds: planar and 3D sliding, rockfall and toppling. The analysis carried out in this study aimed to recognize this variability along the road.

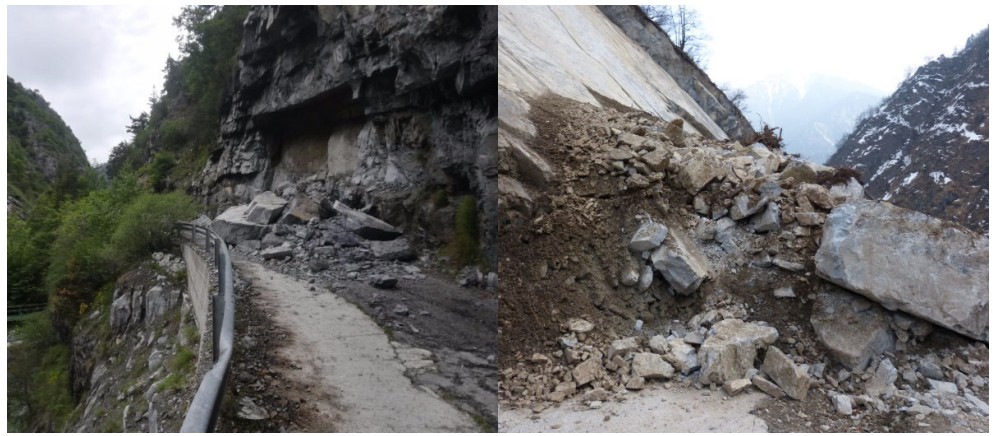

**Figure 4.** Instabilities of 2 June 2018 (about 250 m$^3$) and 25 February 2015 (more than 2000 m$^3$).

**Table 1.** Historical data of the main instability phenomena and related road sections (the position of the sections is indicated in Figures 8 and 9) [31,32].

| Instability Phenomena | | |
|---|---|---|
| **km** | **Date** | **Characteristic of Instability** |
| 0 + 150 | 1996 | • Erosion phenomena at the foot of the retaining wall |
| 0 + 310 | 3 March 2013 | • Slabs up to 1 m$^3$ and rock fragments fell vertically on the road |
| | 3 June 2018 | • Slabs up to 5 m$^3$ underwent probable breakage due to bending at a height of about 8 m |
| 0 + 550 | 1999 | • About 15 m$^3$ of rocky material detached from the slope |
| 0 + 650 | 11 December 2014 | • A landslide occurred with over 20 m$^3$ of stone material |
| 0 + 700 | 13 February 2014 | • Blocks of about 0.5 m$^3$ detached from rock walls with high heights |
| 1 + 400 | 16 February 2013 | • A rocky spur of about 350 m$^3$ slid from the strongly verticalized walls |
| 1 + 650 | 12 December 2015 | • Stone material of about 0.5 m$^3$ was deposited on the road |

**Table 1.** *Cont.*

| km | Date | Characteristic of Instability |
|---|---|---|
| | | **Instability Phenomena** |
| 1 + 850 | 1972 | • An imposing portion of rock of about 1200 m$^3$ slid along the slope |
| | 1999 | • A portion of rock of about 240 m$^3$ slid along the slope |
| | 10 September 2007 | • A portion of rock of about 45 m$^3$ slid along the slope |
| | 28 February 2015 | • An imposing portion of rock of about 2500 m$^3$ slid along the slope |
| 2 + 900 | 10 March 2008 | • Some rocky blocks of about 15 m$^3$ slid along the slope |
| 3 + 000 | 4 April 2016 | • A considerable number of rocky blocks of about 20 m$^3$ collapsed along the slope |

## 3. Methodology, Surveys and Data Analysis

In order to obtain high-resolution images along the road characterized by a complex orography with overhanging and inaccessible rock walls, UAV and terrestrial photogrammetry supported by a GNSS (Global Navigation Satellite System) topographic survey was carried out.

Through the AMS (Agisoft Metashape Professional, vrs. 1.7.4) software [33], a 3D digital model of the entire slope along the road was obtained, processing the data acquired with an aerial and terrestrial photogrammetry survey. As part of the multiscale approach and in order to collect discontinuity data useful for kinematic analysis, another 20 detailed DSMs, with higher point resolutions, of slope sectors were created in areas of particular interest.

The no-contact discontinuity surveys were performed with the Rockscan software [34] developed to facilitate the interaction with the point cloud in the identification of the discontinuities and obtain metrical data of orientation, spacing and trace length. The orientation data obtained from the no-contact survey were statistically processed by the software Dips7.0 [35], obtaining for each sector the stereogram on which the discontinuity set was identified. Through the same software, in each sector, potential kinematic mechanisms were identified considering the stability of the various sectors and the volumes potentially related to the geometric conditions.

During the in situ activities, a rapid compass traditional survey was carried out. This survey cannot be considered either complete or exhaustive, but as a support for indirect measurement due to the high level of risk associated with the unstable slope condition.

The methodologies and software used are explained in more detail in the following chapters.

### 3.1. Topographic and Photogrammetric Surveys

The photogrammetric and topographic surveys were carried out along 9 km of rocky walls overlooking the road. About 2000 photogrammetric images were taken, obtaining very dense 3D point clouds, with more than 2 billion points.

In order to obtain all the metric information regarding the slopes overlooking the ten kilometres of the Elva Valley Road, a 3D integrated metric survey was performed employing a multisensor and multiscale survey with different techniques [36]: UAVs (Unmanned Aerial Vehicle), SLAM techniques (Simultaneous Localizations Additionally, Mapping) and CRP (Close-Range Photogrammetry). Particularly, UAVs and CRP techniques were used to generate DSMs and orthophotos useful for the following slopes analysis.

In order to cover all nine kilometres of the road, the survey was subdivided into three different domains, namely "High part of Road (HR)", "Low part of Road (LR)" and "Large Landslide (LL)", respectively, as shown in Figure 5.

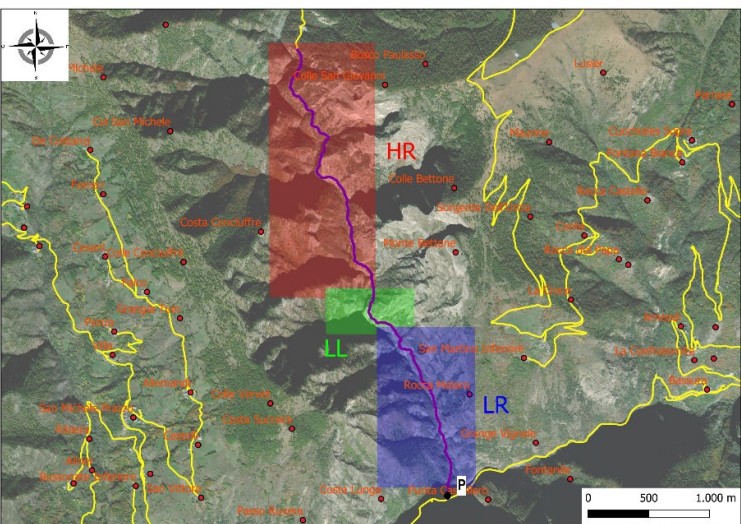

**Figure 5.** Representative surveys domains: High Road (HR), Large Landslide (LL), Low Road (LR), coordinates of point P (UTM zone 32T, 349,027.40 m E, 4,938,557.42 m N).

During UAV surveys, about 1700 digital images were taken in 19 flights by using a commercial drone (DJI Phantom 4, with a FC6310 camera, 20 MP CMOS Complementary Metal Oxyde Semiconductor sensor, focal length of 8.8 mm). In order to perform an appropriate acquisition of the almost vertical surface of the outcropping rock, both nadiral and oblique images were taken using standard criteria of terrain resolution and accuracy [37]. Nadiral and oblique images were combined as shown in Table 2.

**Table 2.** UAV photogrammetric data.

| Area | Nadir | Oblique | Total |
| --- | --- | --- | --- |
| HR | 3 flights, 505 images | 5 flights, 395 images | 8 flights, 900 images |
| LR | 4 flights, 340 images | 3 flights, 259 images | 7 flights, 599 images |
| LL | 2 flights, 48 images | 2 flights, 137 images | 4 flight, 185 images |

The aerial images were combined with high-resolution images taken using the terrestrial photogrammetry technique.

In the process of acquiring terrestrial images, a particular pattern was used: parallel to the rock wall flank and with 45° on the left and on the right along the road with a possible maximum distance from the flanks to optimize the resolution. In this case, more than 3400 digital images were taken along the entire road, using high-resolution cameras (NIKON D800E, 24 mm focal length and 36,3 MP CMOS Full-Frame sensor, and SONY ILCE-7RM3, 24 mm focal length and 42,4 MP CMOS Exmor R Full-Frame 35 mm sensor).

In order to georeferenced every digital model processed, a topographical network was designed. About 200 points were measured by GNSS or total station techniques, guaranteeing a high accuracy and precision ($\sigma_{max}$= 3 mm in vertical or horizontal directions): (i) 22 horizontal markers materialised along the road paving, where the coordinates were estimated considering a multibase solution (through the Leica Geo Office® software v.8.4) with the Ostana and Demonte permanent stations of the CORSs (Continuous Operating Reference Stations) network by the Piedmont district; (ii) 54 vertical markers and 120 natural targets along the rock outcropping (about 1 marker every 50 m) were acquired using total station instrument. In Figure 6, an example of the topographic marker acquisition surveys is shown.

Some of them were considered as Ground Control Points (GCPs) for georeferencing the 3D models, while the others were used as Check Points (CPs) to validate the obtained model precision.

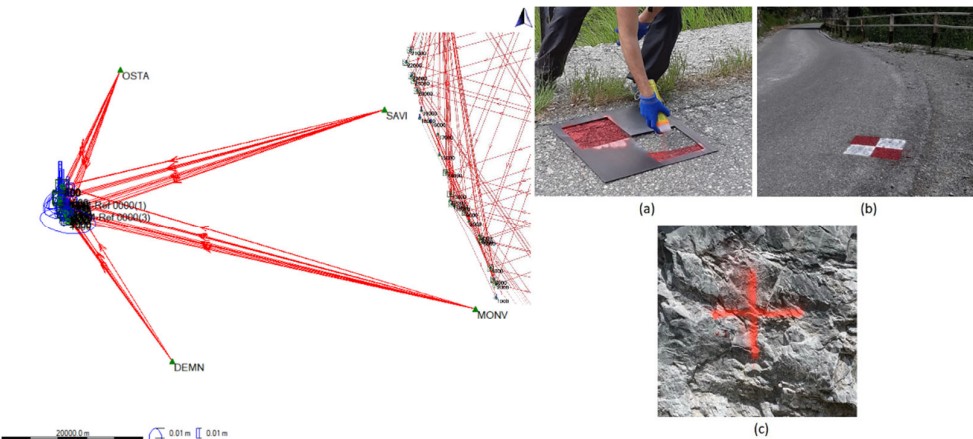

**Figure 6.** CORS constellations in Elva area and an example of the markers materialized along the road paving and rock outcropping: (**a**) colouring of highly visible markers (30 × 30 cm) along the street, using a specially made template; (**b**) a horizontal marker; (**c**) a vertical marker.

### 3.1.1. Photogrammetric Data Processing

The AMS software [33] was used to process the data acquired through aerial and terrestrial photogrammetry, using the implemented SfM (Structure for Motion) technique [17,38–40]. In the multiscale approach applied in this study, the photogrammetric data processing was performed in order to obtain digital models at different resolution scales: UAV data were processed to obtain the DSM of the entire road, the terrestrial images were used to subsequently reconstruct detail DSMs.

### 3.1.2. UAV Data Processing

The imagines obtained by UAV were used to build the DSMs and the relative orthophotos of the whole road using both nadiral and oblique images, setting up the highest level of accuracy in the workflow, i.e., using the photos at the original size, generating high-density point clouds (about 1 point every 4 cm for a total number of about 520 million points) and using 1/5 of the number of points of the sourced dense point clouds for generating high-resolution meshes (Figure 7). To georeference the 3D models and validate the obtained precision, several GCPs and CPs were collimated in all the images, obtaining the results shown in Table 3.

**Table 3.** Number of GCPs and CPs, dense cloud points, and DSM faces, and vertices used and obtained in the different road sections.

| Sector | N° GCPs | N° CPs | Dense Cloud | | DSM | |
|--------|---------|--------|-------------|------|-----|------|
| | | | N° Points [millions] | Density [Points/dm²] | N. Faces [Millions] | N. Vertices [Millions] |
| HR | 17 | 7 | 268 | 6 | 53 | 27 |
| LR | 9 | 4 | 47 | 10 | 0.1 | 0.1 |
| LL | 13 | 5 | 199 | 8 | 40 | 120 |

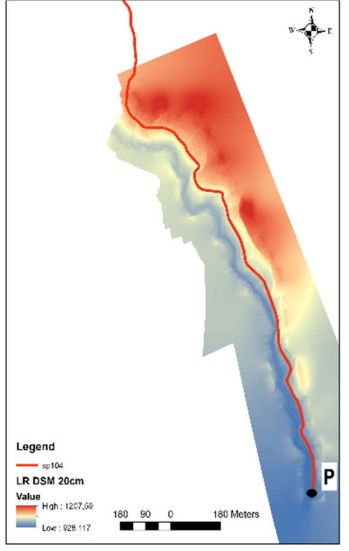 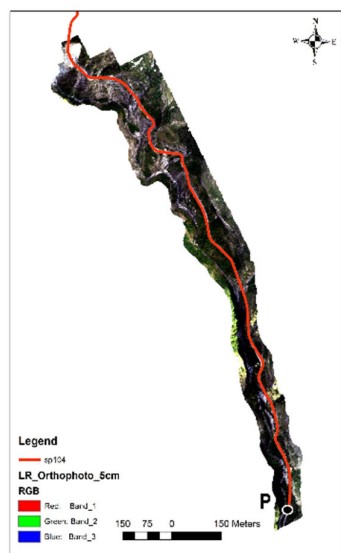

**Figure 7.** LR example of DSM and orthophoto; coordinates of point P (UTM zone 32T, 349,027.40 m E, 4,938,557.42 m N).

### 3.1.3. Terrestrial Data Processing

First, the photogrammetric processing of terrestrial images was conducted with the aim of integrating point clouds obtained through UAV data.

As with UAV photogrammetric data processing, within a multiscale process, the rock wall flanks were reconstructed in a large scale along the whole road, setting the models to a "high" level of accuracy of AMS. In addition, two dense point clouds with a "high" level of detail were generated. Table 4 below summarizes the results obtained in terms of the number of points and density of the two dense clouds.

**Table 4.** Terrestrial photogrammetric dense point clouds statistics.

| Sector | N° Point [Millions] | Density [Points/dm$^2$] |
|--------|---------------------|-------------------------|
| HR | 2668 | 526 |
| LR | 772 | 192 |

Second, starting from dense point clouds and terrestrial images, it was possible to develop detailed models in some sectors along the entire road.

Twenty rock mass outcroppings located along the road were subject to geo-mechanical survey with both quick traditional and no-contact techniques: 11 along the HR, 9 along the LR and 1 at the LL. A detailed DSM was built in correspondence to each of the sites. Figures 8 and 9 at the end of this paper show the orthophotos in order to better visualize the position of all sectors.

To also reproduce the smaller discontinuity surfaces, the DSMs must be characterized by a high density of points, sometimes less than 1 pt/cm$^2$, depending on the area of emergent discontinuity surfaces.

With the general survey, it was not possible to obtain a sufficient number of points, and with this aim, 20 detailed DSMs were built reproducing a portion of the rock slope with limited dimensions: about 30–40 m long and 15 m high; most of the surveyed rock masses did not include any GCPs or CPs, as reported in Table 3; for this reason, each DSM was georeferenced by recognizing and defining several natural points (e.g., markers in Figure 10) on the digital images and by obtaining their coordinates directly from the general DSM of the entire road. As an example, the results for two characteristic sectors considered are shown in Figure 11.

Table 5 summarizes the results obtained in terms of the number of points and density of the dense clouds, number of faces and vertices for each DSM.

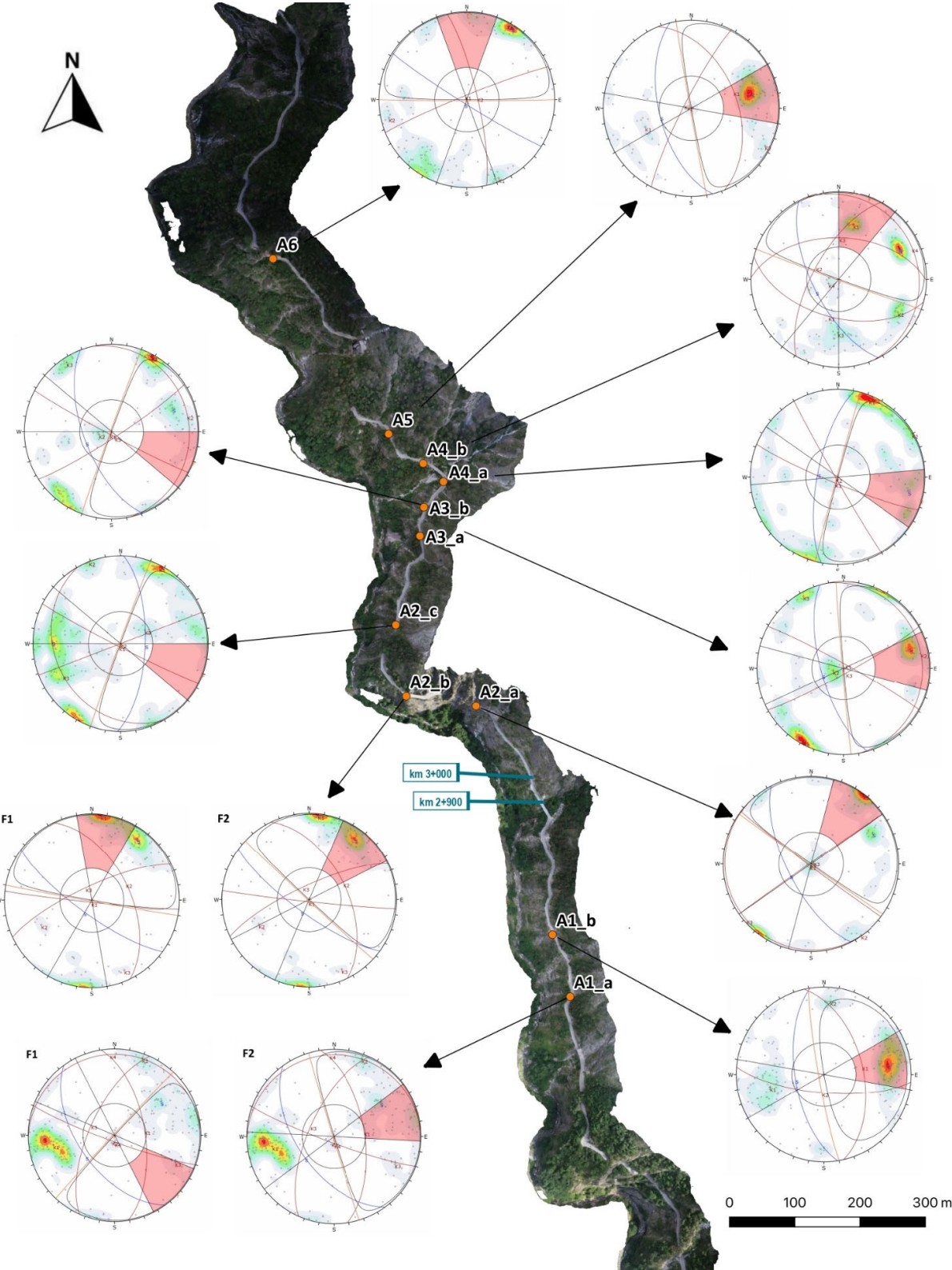

**Figure 8.** Orthophotos and results of kinematic analysis: High Road.

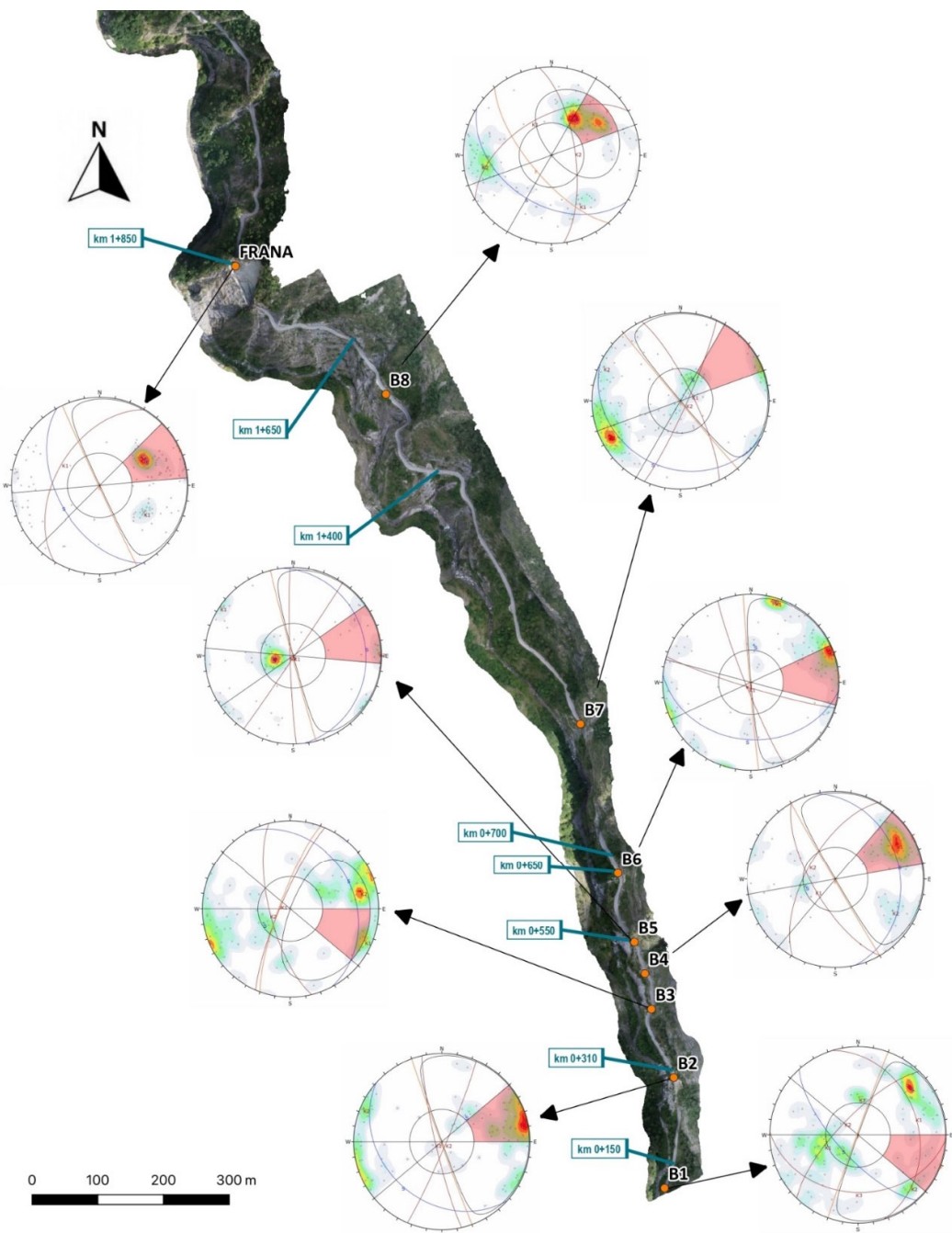

**Figure 9.** Orthophotos and results of kinematic analysis: Low Road and Large Landslide.

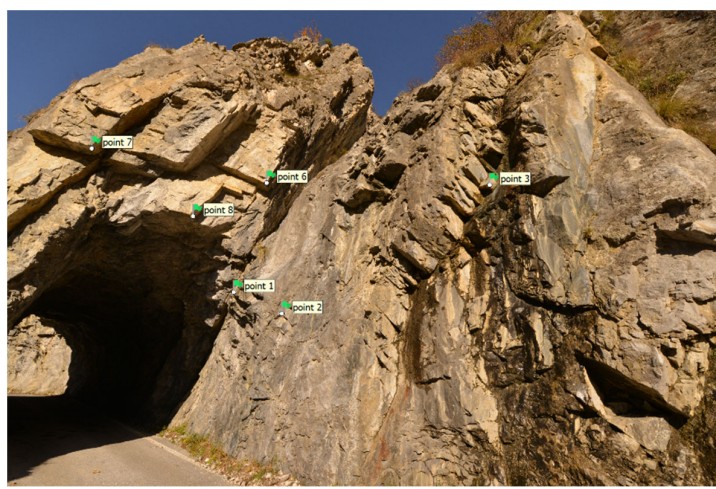

**Figure 10.** Examples of natural markers used for georeferencing the specific DSMs.

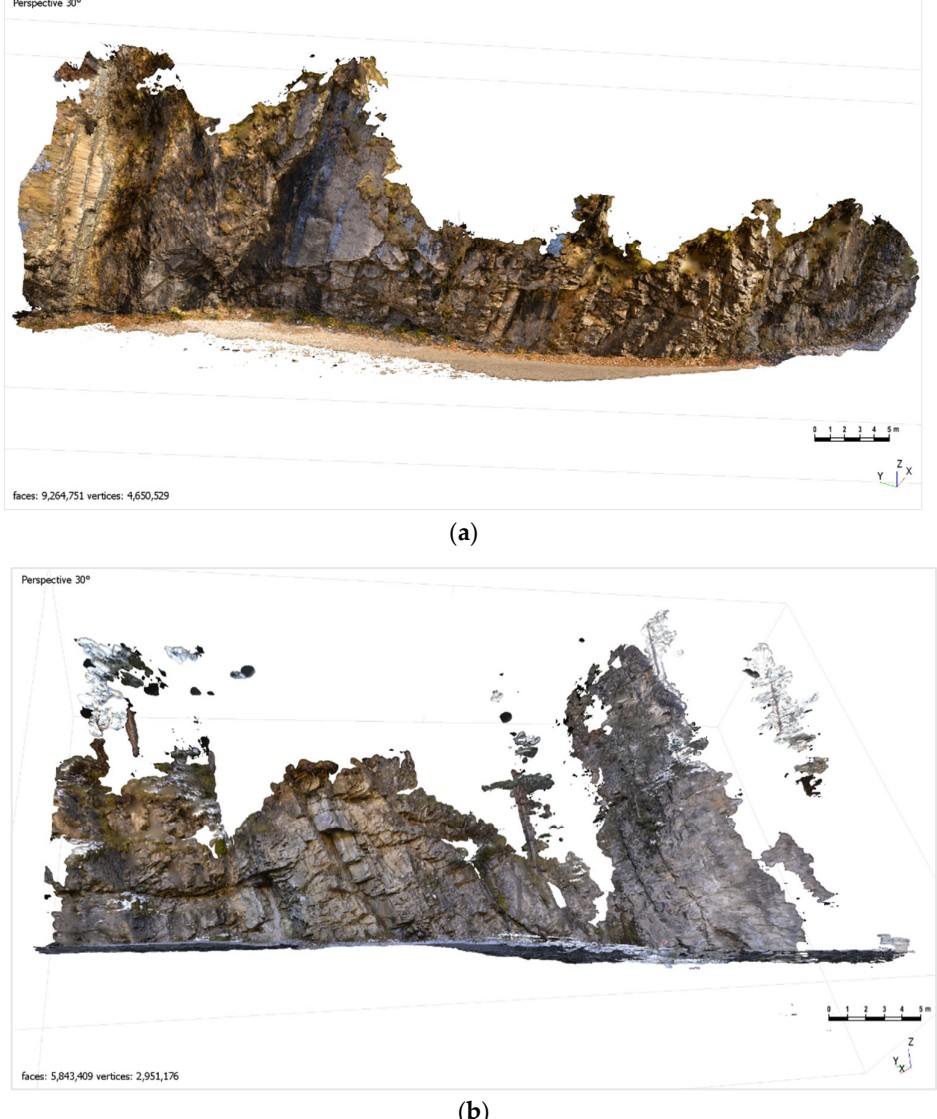

**Figure 11.** Three-dimensional mesh for sectors: (**a**) A1_b and (**b**) A3_b (Figure 8).

**Table 5.** Dense cloud points, DSM faces, and vertices used and obtained for each detailed DSM.

| Sector | Dense Cloud | | DSM | |
|---|---|---|---|---|
| | Points [$10^6$] | Density [Points/dm$^2$] | Faces [$10^6$] | Vertices [$10^6$] |
| B1 | 61 | 1150 | 9 | 4.6 |
| B2 | 52 | 879 | 10.5 | 5.3 |
| B3 | 43 | 783 | 8.6 | 4.3 |
| B4 | 59 | 394 | 11.7 | 5.9 |
| B5 | 77 | 467 | 2.6 | 1.3 |
| B6 | 72 | 562 | 2.8 | 1.4 |
| B7 | 21 | 279 | 1.9 | 0.9 |
| B8 | 53 | 244 | 5.4 | 2.7 |
| LL | 46 | 225 | 0.077 | 0.038 |
| A1_a | 55 | 1573 | 5.8 | 2.9 |
| A1_b | 64 | 915 | 9.3 | 4.6 |
| A2_a | 54 | 1158 | 10.9 | 5.5 |
| A2_b | 86 | 808 | 12.8 | 6.5 |
| A2_c | 58 | 808 | 11.7 | 5.9 |
| A3_a | 35 | 1072 | 7 | 3.5 |
| A3_b | 30 | 986 | 5.8 | 2.9 |
| A4_a | 40 | 778 | 4.8 | 2.4 |
| A4_b | 54 | 952 | 4.1 | 2 |
| A5 | 32 | 1059 | 6.5 | 3.3 |
| A6 | 33 | 1854 | 6.7 | 3.4 |

*3.2. No-Contact Survey Tools—Rockscan*

The DSMs, the oriented digital images and the calibration parameters of the camera were used in the Rockscan software [34], specifically developed for measuring the orientation and spacing of the plane (Figure 12).

This program calculates the orientation and position of the discontinuity surfaces, with an interface that, through the image of the wall itself, allows the visual recognition and selection of discontinuity planes. The selection of the regions of interest (ROIs) on the photograph corresponds to the selection of the relative portion of DSM and, therefore, to a certain number of oriented points, belonging to the surface of the selected wall.

Through the RANdom SAmple Consensus (RANSAC) [41] procedure of geometric segmentation of the DSM, it is possible to determine the plane or planes that better approximate the geometry and to define the orientation, in terms of dip and dip direction, as well as the coordinates of the barycentre of the plane or planes identified.

Furthermore, it was possible to carry out spacing measurements between the joints belonging to the same discontinuity system. The possibility of making distance measurements is equally useful in situations where it is necessary to identify the dimensions of unstable blocks or depletions (Figure 13).

In this way, with this software, the 20 portions of rock mass were analysed, and the relevant data were obtained. More than 2560 plane orientation datasets and 1000 spacing measurements were collected, allowing the authors to perform kinematic analyses to evaluate how the degree of stability of the rock mass changes along the way.

For each sector, a stereographic projection of the collected plans made it possible to identify the systems of discontinuities and the orientation of their main plan; the spacing data were statistically analysed obtaining frequency curves and their characteristic values (maximum, minimum, average, modal value, etc.).

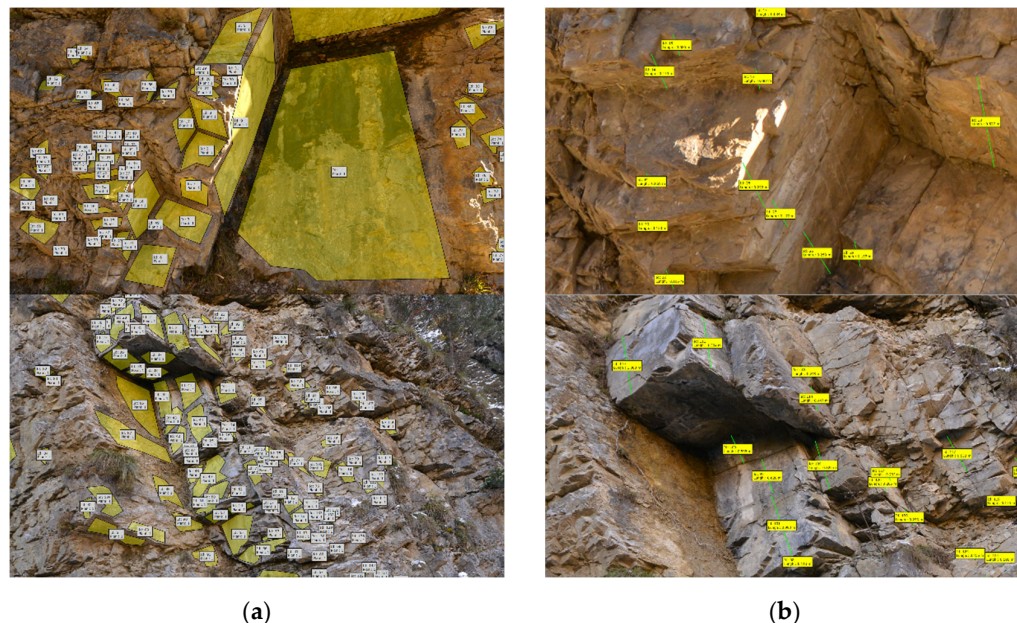

**Figure 12.** Discontinuity data collected by Rockscan software in portions of A1_b (above) and A3_b (below) sectors: (**a**) discontinuity planes and (**b**) spacing.

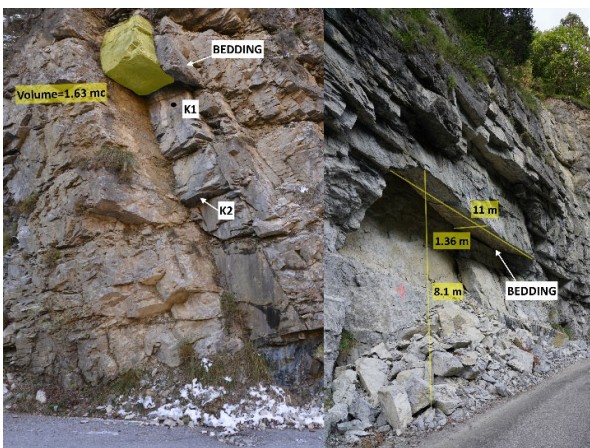

**Figure 13.** An example of distance measurements of unstable block and depletion.

### 3.3. Kinematics Analysis

Potential kinematic mechanisms and the volume of blocks potentially involved were identified in each sector. The analysis was performed by Dips 7.0 [35] considering the stability of the various sectors only in relation to the geometric conditions, excluding potential discontinuity resistance parameters or the external aging forces (hydrostatic thrust, pseudo-static earthquake action, reinforcement and/or stabilization systems). In fact, this kinematic approach only identifies potential failure mechanisms, so a detailed stability analysis with conveniently estimated parameters should be then necessary to quantify the actual stability condition in each analysed sector.

The identification of the kinematic mechanisms that can occur is possible thanks to the application of the Markland test [42]. This allows the identification of the type or types of instability (planar and wedge sliding, toppling, and rock fall) thanks to the comparison of the discontinuity system orientation to the slope one.

To estimate the volume of the blocks involved in each sector, the analytical equation proposed by Palmstrøm [43], whereby in the presence of three families of discontinuities, was used:

$$V_b = (S_1 \cdot S_2 \cdot S_3)/(\text{sen}\gamma_1 \cdot \text{sen}\gamma_2 \cdot \text{sen}\gamma_3) \tag{1}$$

where, $S_1$, $S_2$, and $S_3$ represent the spacings of the relative discontinuity systems, and $\gamma_1$, $\gamma_2$, and $\gamma_3$, the angles between the systems.

Considering the statistical variability of the spacing values, it is possible to obtain a frequency distribution of blocks volumes (in situ block size distribution) [44,45]. In this study, a combination of maximum, minimum, average and modal spacing values of each discontinuity system was considered to obtain an indication of the volume range.

## 4. Results

The identification of the discontinuity plans and the following kinematic analysis were carried out for each of the 20 sectors considered. In Figure 14, stereograms for sectors A1_b and A3_b are reported, while Table 6 indicates, for each sector, the number of planes detected and the mean orientation data of the discontinuity systems identified. In each sector, three main systems were identified in accordance with the geological evidence: a bedding planes (BP), the most persistent, and two conjugated systems (K1 and K2) perpendicular to the bedding plane with a limited persistence close to BP spacing. In some sectors, where the rock is more fractured, two other systems, K3 and K4, were identified.

Spacing data were analysed, system by system, through a frequency distribution analysis (Table 6), organizing the data into classes, as indicated in the ISRM [2]: less than 2 cm; from 2 to 6 cm; from 6 to 20 cm; from 20 to 60 cm; from 60 to 200 cm; from 200 to 600 cm and greater than 600 cm. Figure 15 reports the spacing frequency distribution and the cumulative frequency curve obtained for the sectors A1_b and A3_b, used as examples.

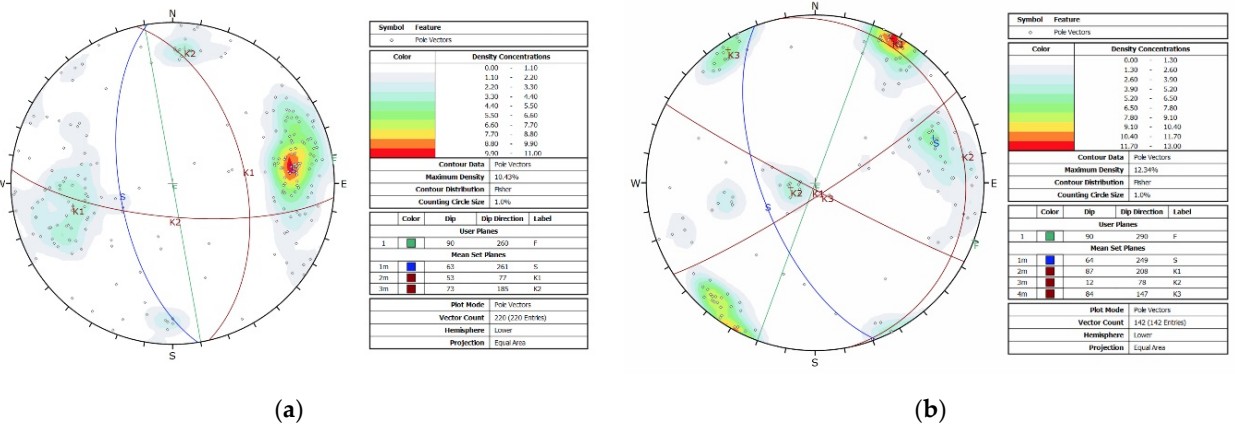

(**a**)           (**b**)

**Figure 14.** Stereogram for sectors (**a**) A1_b and (**b**) A3_b.

**Table 6.** Number of planes surveyed for each sector; mean orientation data and average spacing values for each discontinuity set.

| Sector | Total N. of Planes | Mean Orientation of Discontinuity Set | | | | | Average Spacing for Each Set [cm] | | | | |
|---|---|---|---|---|---|---|---|---|---|---|---|
| | | BP | K1 | K2 | K3 | K4 | BP | K1 | K2 | K3 | K4 |
| | | DipDir;Dip [°] | DipDir;Dip [°] | DipDir;Dip [°] | DipDir;Dip [°] | DipDir;Dip [°] | | | | | |
| B1 | 106 | 228;70 | 073;32 | 314;73 | 182;35 | / | 14 | / | / | / | / |
| B2 | 97 | 222;33 | 257;84 | 113;86 | / | / | 71 | / | / | / | / |
| B3 | 87 | 062;28 | 293;81 | 256;71 | / | / | 79 | / | / | / | / |
| B4 | 96 | 077;28 | 299;63 | 240;68 | / | / | 105 | 34 | / | / | / |
| B5 | 92 | 083;19 | 124;89 | 268;87 | / | / | 108 | 77 | / | / | / |
| B6 | 118 | 185;35 | 197;85 | 248;85 | / | / | 115 | / | / | / | / |
| B7 | 166 | 205;24 | 078;64 | 114;84 | / | / | 51 | / | / | / | / |
| B8 | 150 | 212;43 | 330;54 | 82;68 | / | / | 22 | / | / | / | / |
| LL | 157 | 241;50 | 301;49 | / | / | / | 51 | / | / | / | / |
| A1_a | 241 | 234;56 | 081;61 | 200;84 | 294;66 | 357;07 | 18 | / | 28 | / | 17 |
| A1_b | 220 | 261;63 | 077;53 | 185;73 | / | / | 29 | 27 | 36 | / | / |
| A2_a | 74 | 243;68 | 217;88 | 325;89 | 229;3 | / | 66 | 19 | / | 37 | / |
| A2_b | 116 | 216;77 | 186;88 | 064;54 | 335;77 | / | 47 | / | / | / | / |
| A2_c | 145 | 092;67 | 207;87 | 159;90 | 241;26 | / | 62 | 53 | 68 | / | / |
| A3_a | 97 | 254;69 | 086;29 | 079;10 | 151;85 | / | 23 | 24 | 40 | / | / |
| A3_b | 142 | 249;64 | 208;87 | 078;12 | 147;84 | / | 38 | 46 | 41 | / | / |
| A4_a | 83 | 281;71 | 202;84 | 240;88 | / | / | 38 | / | / | / | / |
| A4_b | 121 | 243;66 | 195;54 | 299;66 | 001;51 | 067;10 | 23 | / | / | / | / |
| A5 | 110 | 256;60 | 068;47 | 297;83 | / | / | 17 | 32 | / | / | / |
| A6 | 133 | 212;86 | 338;86 | 077;80 | / | / | 12 | 20 | / | / | / |

The symbol "/" means that there is no data available.

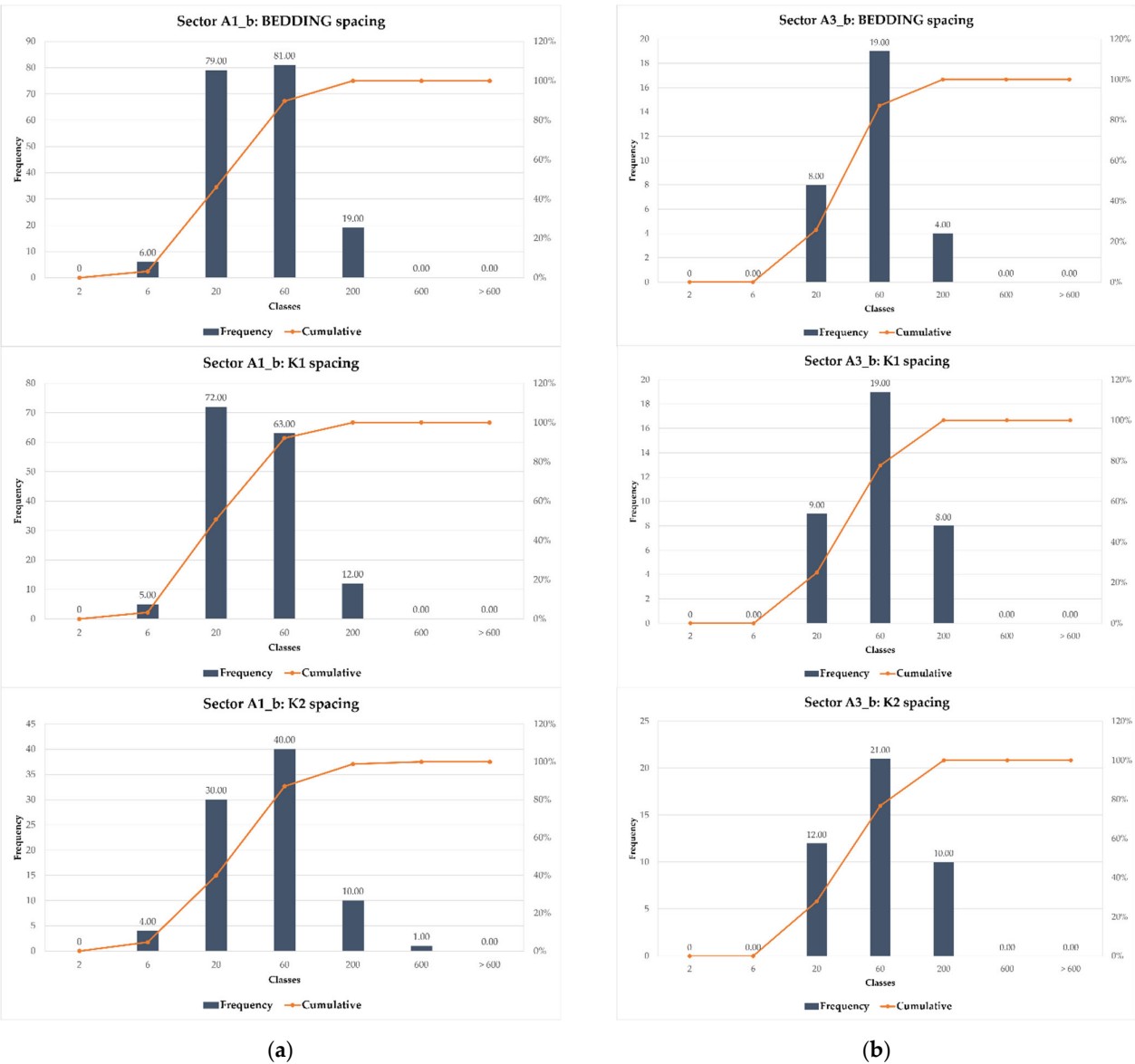

**Figure 15.** Frequency diagrams of the spacing data for sectors (**a**) A1_b and (**b**) A3_b.

In each sector, the potential kinematic mechanisms were identified. As previously indicated, the analyses were performed by Dips7.0 [35], identifying the kinematic mechanisms that can occur, on the base of the planes and the slope orientation through the application of the Markland test [43]. Figure 16 shows the Markland test carried out for the A1_b and A3_b sectors; in Table 7 and in Figures 16 and 17, the results obtained in each sector are summarized. The prevailing kinematic mechanism is that of planar sliding but, in some cases, the discontinuity systems can form blocks, which can become unstable due to rockfall or toppling.

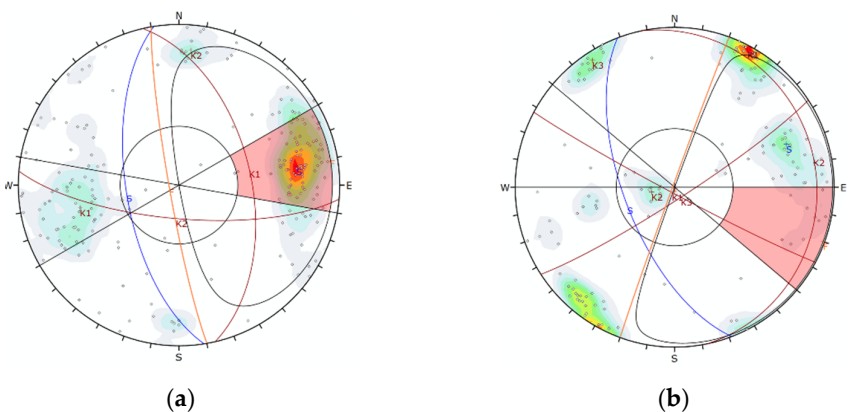

(**a**) (**b**)

**Figure 16.** Markland test with Dips 7.0 for sectors (**a**) A1_b and (**b**) A3_b.

**Table 7.** Kinematic mechanisms identified in each sector (F is the slope face).

| Sector | Kinematic Mechanisms |
|--------|----------------------|
| B1 | • F(290;89): Undercut rock and debris fall vertically due to a lack of support at the foot, triggering the sliding along the BP of pseudo-vertical slabs, if these are free from other systems of discontinuity. |
| B2 | • F(250;90): Planar sliding along K1. Horizontal slabs formed by K1 and BP, free from K2, which can suffer breakage due to bending. |
| B3 | • F(290;80): Horizontal slabs formed by K1 and BP, free from K2, which can suffer breakage due to bending. |
| B4 | • F(240;89): Planar sliding along K1. Horizontal slabs formed by K1 and BP, free from K2, which can suffer breakage due to bending. |
| B5 | • F(255;89): Undercut rock and debris fall vertically due to a lack of support at the foot; fall of blocks from above; breakage can occur due to shearing along joints or within the rock matrix. |
| B6 | • F(265;89): Undercut rock and debris fall vertically due to a lack of support at the foot; fall of blocks from above due to exceeding the tensile strength of the joint. |
| B7 | • F(230;89): Undercut rock and debris fall vertically due to a lack of support at the foot, fall or topple of blocks. |
| B8 | • F(230;70): Planar sliding along BP. Fall of blocks formed by K1 and K2. |
| LL | • F(245;89): Planar sliding along BP. |
| A1_a | • F(313;89): Fall of blocks formed by K1 and K2. Presence of two faults.<br>• F(253;89): Planar sliding along BP. Fall of blocks formed by K2 and K4. |
| A1_b | • F(260;85): Planar sliding along BP. Fall of blocks formed by K1 and K2. |
| A2_a | • F(215;89): Planar sliding along BP. Fall of blocks formed by K1 and K2. |
| A2_b | • F(190;89): Undercut rock and debris fall vertically due to a lack of support at the foot; fall or topple of blocks.<br>• F(223;89): Planar sliding along BP. Undermining of the foot, fall or topple of blocks. |
| A2_c | • F(290;89): Fall of blocks formed by K1 and K2. |
| A3_a | • F(265;89): Planar sliding along BP. Fall of blocks formed by K1 and K2. |
| A3_b | • F(290;89): Planar sliding along BP. Fall of blocks formed by K1 and K2. |

**Table 7.** *Cont.*

| Sector | Kinematic Mechanisms |
|--------|----------------------|
| A4_a | • F(190;89): Planar sliding along BP. Fall of blocks formed by K1 and K2. |
| A4_b | • F(200;89): Planar sliding along K1 |
| A5 | • F(260;89): Planar sliding along BP. |
| A6 | • F(180;89): Planar sliding along BP. Presence of a fault. |

A frequency distribution of the block volumes was obtained by the Palmstrøm [44] equation, considering only the presence of the three principal discontinuity systems (BP, K1 and K2) and a combination of their spacing statistical parameters: maximum, minimum, average and modal values. In each sector, 60 volume values were evaluated, and the frequency distribution and the cumulative distribution obtained in the A1_b and A3_b sectors are shown in Figure 17. The volume of the blocks defined as such is highly variable and ranges from 10 cm$^3$ to 40 m$^3$. In this case, the classes considered for the frequency analysis are: less than 10 cm$^3$; from 10 to 200 cm$^3$; from 0.2 to 10 dm$^3$; from 10 to 200 dm$^3$; from 0.2 to 10 m$^3$ and greater than 10 m$^3$. Table 8 summarizes the results in terms of volume values.

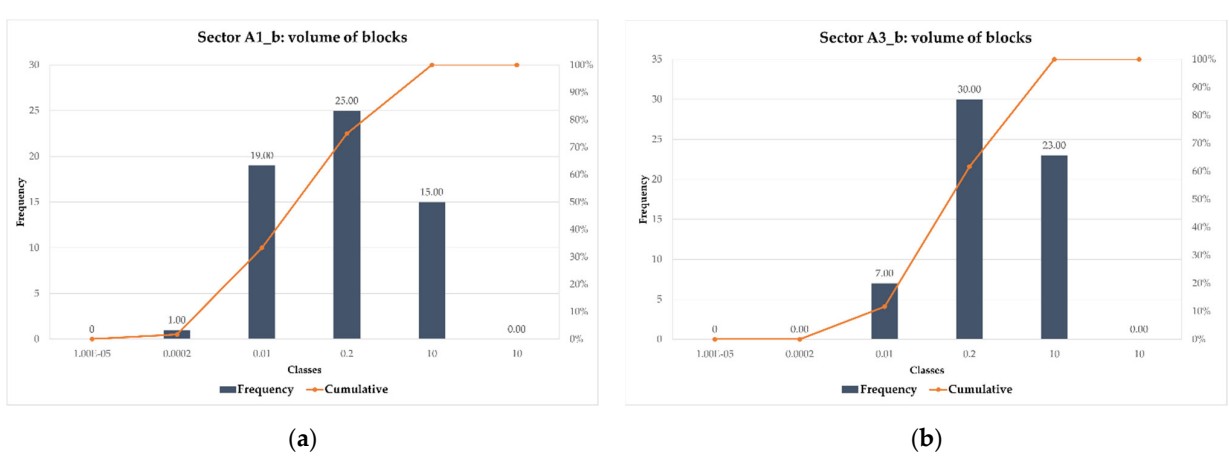

(**a**)           (**b**)

**Figure 17.** Frequency diagrams of volume distributions for sectors (**a**) A1_b and (**b**) A3_b.

**Table 8.** Block volume.

| Sector | | A1_a | A1_b | A2_a | A2_c | A3_a | A3_b |
|--------|---------|------|------|------|------|------|------|
| Block | Average | 0.06 | 0.38 | 0.24 | 2.30 | 0.27 | 0.38 |
| Volume | Maximum | 0.90 | 8.20 | 2.60 | 40 | 6.15 | 4.30 |
| [m$^3$] | Minimum | 0.00012 | 0.000045 | 0.00052 | 0.00061 | 0.000072 | 0.00071 |

## 5. Discussions

Generally, in all the sectors analysed, three discontinuity systems were detected: the bedding plane, which is the most persistent, and at least two more conjugated systems, contained between two continuous bedding planes, with limited persistence close to bedding plane spacing. In most cases, these planes were perpendicular to each other and to the bedding plane, which can form mainly prismatic blocks with different sizes.

The road is winding and follows the course of the river that lies at the foot of the slope with rock walls facing south, west and north. This morphological condition and the presence of faults and folds that change the bedding plane and its conjugated system orientation lead a different slope stability condition along the road.

Figures 8 and 9 summarize the kinematics identified in the sectors, together with the estimate of the volume of the blocks. It is important to underline that the kinematics identified concern the individual blocks, but it is possible that these will take place as a combination of several blocks. Generally, for the case under examination, the prevailing kinematic mechanism is that of planar sliding but, in some cases, the discontinuity systems can form blocks that can become unstable due to collapse or overturning. In these cases, and where it was possible according to the spacing measures, the distribution of the volume of the blocks was defined.

### 5.1. Low Road

In the southern part of the road (LR), the bedding plane is mainly pseudo-horizontal. In fact, except for the first stretch of road, where the BP is inclined at 70°, this plane has a dip that varies between 19° and 35°.

Analysing the spacing data, it can be noted that in the lower part of the road, at least in the first sector (sector B1, Figure 9), the average spacing of the bedding plane is between 80 and 115 cm, forming thick rock banks. This thickness decreases in the northern part of the low road, reaching an average spacing between 20 and 50 cm.

In the lower part of the road, the bedding plane is pseudo-horizontal, and two pseudo vertical conjugate sets, which form horizontal slabs with the bedding plane, can be observed. These slabs can suffer breakage due to bending. In this part of the road, undercut rock and debris fall vertically due to a lack of support at the foot, the mobilization of which is attributable to a break due to the cutting of joints, caused by excessive loads, or even within the rocky matrix. In fact, in some sectors, unstable blocks are clearly visible, which can mobilize due to an increase in the load on the wall, to atmospheric agents or to anthropogenic actions. For the whole part of low road, there are outcropping and overhanging rock masses with a strongly fractured portion projecting onto the road. The rock walls have heights of 20–30 m and, in some places (sector B5, Figure 9), have a "canopy" morphology on the stock (footprint of at least 120 m$^2$).

### 5.2. Large Landslide

Following the valley northwards, the BP begins to verticalize. This is clearly visible in the sector that goes from sector B8 to the LL, where the bedding plane assumes an inclination between 45 and 50° and an average spacing of 50 cm. A planar sliding kinematic mechanism can be triggered along this plane. Furthermore, the presence of other discontinuity systems means that portions of rock of varying shape and size (from small blocks to large slabs) can slide along the bedding plane.

The geometry of the road, connected with the natural stratification of the rock face, gives a dip slope morphology with the presence of disharmonious folds. Since the construction of the road and its widening, this site has exhibited phenomena of planar sliding that have progressively mobilized slabs or portions of slabs released at the foot precisely from the re-profiling of the slope made to allow the construction of the roadway. The natural slope, which was initially characterized by a morphological surface similar to the current lateral slopes that delimit the area of instability, has undergone progressive collapses of considerable magnitude that have resulted in its current morphology (Figure 18).



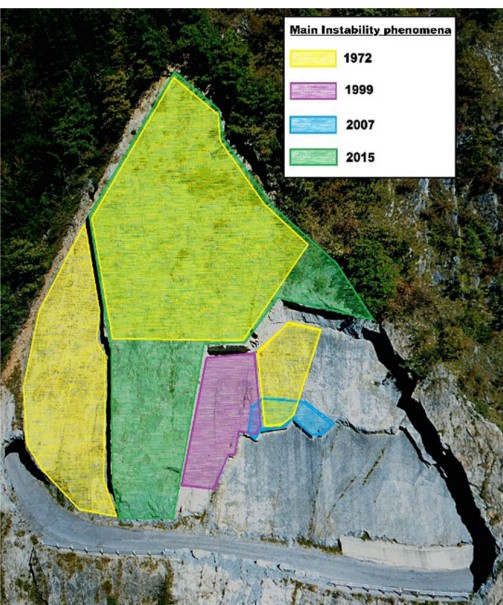

**Figure 18.** Chronology of the collapses in the LL sector (photo 2002).

*5.3. High Road*

Continuing north, in the HR sectors, there are large medium-fractured rock walls arranged with highly inclined stratification (on average 75°), with outcrops of up to 250 m high. In fact, the inclination of BP starts at an angle of 56° and finishes at 86°, and the average spacing varies from 10 to 60 cm. The other two main system, K1 and K2, consequently, change their inclination along the valley.

In the high part of the road, the planar sliding kinematic mechanism can also be triggered along the bedding plane. In this area, the cluster is lower than that in the lower part, but in this part, the discontinuity systems also form small and large blocks that can slide along the bedding plane. Along the entire stretch of road between sectors A3_a and A6 (Figure 8), the stratification of the carbonate rocks becomes vertical with a succession of important watersheds that make the road subject not only to continuous rockfalls, but also to important torrential phenomena and avalanches during the winter.

**6. Conclusions**

The study presented a particularly useful methodology in cases such as that of the Vallone di Elva, in an area with complex orography and many inaccessible parts. It was possible to obtain a diffused characterization with relevant and statistically valid data.

The most advanced survey techniques and methods were adopted, in order to gain a detailed analysis of landslide hazard and the stability condition of rocky slopes.

The main steps of this method are summarized below:

1.  Topographic and photogrammetric acquisition of data: this was conducted in order to obtain all the metric information, markers and natural points to georeference the models, with different techniques—UAV and CRP.
2.  UAV photogrammetric technique: this was used to build the DSMs and the relative orthophoto.
3.  Terrestrial photogrammetric technique: this was used to obtain greater accuracy in terms of the number of points and density.
4.  Creation of detailed DSMs: this was used to obtain DSMs with a high density of points (less than 1 pt/cm$^2$) and to reproduce the smaller discontinuity surfaces.
5.  Interpretation of the no-contact survey:

     a.   Identification of the orientation, position and spacing of the discontinuity surfaces through visual recognition and selection of discontinuity planes (Rockscan, [34]);

b.     Statistical interpretation of the results (Dips7.0, [35]);

c.     Identification potential kinematic mechanisms in relation to geometric conditions.

The application of the methodology to the example under study highlights the advantages of using the indirect survey—more accurate and extended to larger portions of rock—compared to the traditional survey, which is always localized to smaller and, in any case, necessarily accessible portions of rock to the operator. Furthermore, the proposed results show that the combination of UAV and terrestrial photogrammetry techniques made it possible to obtain an even more detailed overview that was necessary to obtain statistically representative data of the entire slope.

The multiscale approach presented in this study was highly useful in the collection of relevant information to analyse how the rock mass geo-structural features can affect the stability slope condition along very long and complex infrastructures.

**Author Contributions:** Conceptualization, M.M. and M.T.C.; methodology, M.M. and M.T.C.; software, A.L., E.P. and M.T.C.; validation, M.M., M.T.C. and C.S.; formal analysis, M.M. and M.T.C.; investigation, A.L. and E.P.; resources, A.L. and E.P.; data curation, A.L. and E.P.; writing—original draft preparation, M.M. and M.T.C.; writing—review and editing, M.M. and M.T.C.; visualization, M.M. and M.T.C.; supervision, C.S.; project administration, M.M.; funding acquisition, C.S. All authors have read and agreed to the published version of the manuscript.

**Funding:** This research received no external funding.

**Institutional Review Board Statement:** Not applicable.

**Informed Consent Statement:** Not applicable.

**Data Availability Statement:** Not applicable.

**Conflicts of Interest:** The authors declare no conflict of interest.

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
