# Peer review of "Rock Mass Characterization by UAV and Close-Range Photogrammetry: A Multiscale Approach Applied along the Vallone dell’Elva Road (Italy)"

_geosciences, doi:10.3390/geosciences11110436_

Round 1

Reviewer 1 Report

In this manuscript, the authors present a study where they quantified rock mass geometric parameters in a 9 km long mountain road partially based on geometry data derived from point clouds remotely acquired. They use this relevant information in order to study the influence on stability of various geostructures at different scales and, ultimately, to analyze road stability by identifying areas susceptible to different failure mechanisms. They do not perform though, proper rock slope stability analysis, something that may be deserved to be mentioned in the introduction to clearly state the scope of the study.

To the best knowledge of this reviewer this is an interesting case study contributing to advance towards a better application of advance surveying techniques to rock slope stability analysis in road geomechanics. So this reviewer recommends publication of this interesting and well explained case study.

However, some minor issues are mentioned below with the aim of improving authors’ work.

In line 171 there are two points together. Remove one. Also the authors use the acronym GNSS, please explain the meaning.

In Figures 8 and 9 it is difficult to see anything, consider remove them.

In page 10, the authors present 20 detailed DTMs that have built reproducing portion of rock slope. Please justify in more detail manner why this is necessary in the context of the general approach to study stability issues in the Elva valley road.

In line 296-298 the authors write “In fact, the software allows 296 the operator to insert one or more digital photographs of the slope, taken during the photogrammetric survey phases, appropriately projecting the DSM of the wall on them its selves.” This is unclear, please re-write.

In lines 307 to 308 there is a jump in the text. Please, correct.

Lines 315-316: “It can be noted that selecting a plane in an image this is 315 also projected in the other frames in which it is visible.” Unclear, please re-write.

In line 329 change “More than 2560 plan” to “More than 2560 plane”. In lines 332 and 333 change “plan” to “planes”. In 337 plans to planes.

On line 341 change “discontinuity systems” to “discontinuity or joint sets”.

Please indicate the criteria to order the joint sets K1, K2… in each sector. Is this just base in pole density or other geological criterion is accounted for?

In line 348 change “contents” to “contained”.

In line 360 include analysis after “through a frequency distribution”.

Based on the changes in orientations of the main joint sets, as for instance in table 5, some comments on the relation of the changing orientations in the context of the structural geology in the area would be welcome. A paragraph would suffice.

In Figure 15, please explain the actual meaning of the classes shown in the histograms. For instance, does 60 means between 40 and 130 cms???? Please, do the same for Figure 17 referring to block volume.

In lines 396 to 398 the authors state:” In fact, it has been not performed a real statistical analysis, but it has been considered the variability of the spacing data in order to determine the volume one’s.”. This is unclear, please re-write.

In line 414 use summarize instead of “summarizes”

In line 432 “In the part of the road too” is unclear, re-write. Preferably, the whole sentence.

In line 463-466, the sentence is unclear, please re-write.

In section 6 the authors propose a kinematic analysis, which is indicative on potential failure mechanisms. However, they do not mention at all geotechnical parameters such as friction or roughness of the joints and so on. The authors should either comment on some basic rock mechanics parameters or indicate that the kinematic approach only identifies potential failure mechanisms, so a detailed stability analysis with conveniently estimated parameters is then necessary to quantify the actual stability of the different zones under scrutiny.

Reference 29 and 51 are the same. Remove the last one and update the text.

Reviewer 2 Report

Minor

The abstract is something light and does not give a complete picture of the content

line 27. …discontinuities geometrical features. Perhaps is better “mechanical discontinuities”

line 28. …rule. Perhaps is better control

line 29. …safety conditions. Perhaps is slope stabilization

line 44. Sometimes. This word is very ambigous in this sentence. Scanline technique (Priest, 1993) is an accepted technique with limitations, but evidently need the accessibility of the user.

Priest, S. D., 1993, Discontinuity analysis for rock engineering: London, Chapman & Hall, 473 p.

lines 46-49. This sentence is weak to explain the inappropiate of the scanline. It is evidently that scanline can be used in accesible places.

lines 51-53. This sentence is not clear. Data capture by photogrammetric or LIDAR (LIdar Detection And Ranging techniques allow apply process to measure …..

lines 51. laser scanner is a device, the technique is LIDAR. Method is a process of a technique. In the sentence, it is ambiguous.

The references are not updated neither sufficient representative in the introduction, also are not detailed

Lines 55-56. …the data obtained from the survey provide direct measurements of the orientation and position of emerging discontinuities. What mean direct measurements? The techniques capture points and a process transform the data in orientation or position of the feature, after a detection, clustering and classification.

Line 57. …These techniques.. What techniques, to capture or process measures, it is confuse.

Line 60….these innovative techniques  (line 57, these techniques have benn used since the end of the last century…). This is contradictory, and the reader may not know what techniques the authors are referring to.

line 62.. fracturing. The authors introduce discontinuity geometrical features, discontinuities. It’s required a introduction for these features, because it is the main feature of the manuscript. Definition of join, if this structure is the same that a fracture or mechanical discontinuity…

line 79. … Due to the numerous instability phenomena occurred since its construction. Can be the information expanded? Figures, table or references

Figure 2. Insufficient information. The figure require of position and view of sight to place the readers.

Figure 3.  Insufficient information. The figure require of North, scale, coordinates…

line 104. …currently active…It is a weak information, can be more detailed?

lines 112-117. The information provided is very vague and imprecise. I think is required an approximate frequency and volumes because is cited in lines 120-123. What is the source of this information?

line 120. The phenomena.If this phenomena is referred to rockfall must be cited.

lines 124-128. This chapter is The Vallone dell’Elva road. Then, the sentence don’t must be here.

line 133. Survey’s technique. The introduction to this chapter is repeated in part in the introduction and the rest must be included in the introduction.

Line 160 DTMs. The abstract is referred to DSM.

Line 171. The acronyms GNSS, is not introduced.

Line 173 DSM is referred, but is confused the acuisition of this model.

Figure 5.  Insufficient information. The figure require of North, scale, coordinates…

Line 194. The acronyms MP, CMOS are not introduced.

Line 226.  using the implemented SfM algorithm. This sentence is not correct, Agisoft use the SfM technique based in a collection of algorithms like SURF or SIFT. More representative references can be:

Snavely, N., Seitz, S., Szeliski, R., 2008. Modeling the World from Internet Photo Collections. International Journal of Computer Vision, 80(2), 189-210. https://doi.org/10.1007/s11263-007-0107-3

Westoby, M.J., Brasington, J., Glasser, N.F., Hambrey, M.J., and Reynolds, J.M., 2012. “Structure-from-Motion” photogrammetry: A low-cost, effective tool for geoscience applications. Geomorphology,  179, 300-314.https://doi.org/10.1016/j.geomorph.2012.08.021

Figure 7.  Insufficient information. The figure require of, coordinates…

Line 250. What are the reasons for assigning a 1:200 scale. Reolution is not proportional to scale. What are the errors of the DSMs or DTMs?

Figure 7.  Insufficient information. The figure require of coordinates…

Figure 8. Strada ala, strada bassa, High, Low road sections,High road, LR, HR too much denominations for the same. Figure 8b is totally useless.

Figure 9. What are the reasons for assigning a 1:5000 scale. The symbols are not visibles. What is the mean of the symbols?

Line 288. Is it correct to  use the term paragraph?

Line 294. …allows automatic identification of the orientation and… The program calculates the orientation of selected points.  Automatic identification is not a synomym of calculate.

Line 304 Through appropriate processes of geometric segmentation. The correponding  processes must be cited.

Line 342. The following Table 5 indicates for each sector the number of planes detected and the orientation data of the families of joints identified. Sector is referred before as a section?  orientation data of the fracture sets is mean orientaion data?.  …family Is supossed to referred to fracture set (Priest, 1993), which must be introduced.

Priest, S.D., 1993. Discontinuity analysis for rock engineering. London,  Chapman & Hall, 473pp. https://doi.org/10.1007/978-94-011-1498-1

Table 5 shows an average of joint spacing and the first item is referred Bedding. What is meaning? Spacing is referred to joints, bedding is referred to layer thickness. The authors uses discontinity family but before is used joint and the table use K1, K2, K3, K4. The confusion is important.

MAJOR

The abstract does not reflect the content of the manuscript. It is focused on the methodology, which is nothing new, using commercial and well-known software.

The structure of the article does not respect the recommendations of the journal Introduction, Methodology, Results, Discussion and Conclusions. The structure presented by the authors (Introduction, The Vallones dell'Elva road, Survey's techniques, Topographic and photogrammetric surveys, No-contact discontinuity survey, Kinematics analysis and conclusions) does not improve these recommendations and mixes many concepts in different sections making reading difficult from the article. Some chapters are very long without contributing much. The terminology used is incorrect many times and is not unified. The figures are formally incorrect and wanting to cover the wide study area, they do not illustrate what they are intended to illustrate. The references used are not very representative in some topics. At no time is there a mention of the errors obtained in the elaboration of the digital terrain models.

An important part of the manuscript is focused on the methodology, which does not present any novelty and could be much more reduced. The discussion and results, which in my opinion is the contribution of the article, is not well explained and little helped with the figures.

The article explains an extensive methodological work to briefly show the results and discussions of the studied area.

Round 2

Reviewer 2 Report

Minor

Line 15. no-contact (But the authors explain  that they have made  manual scanlines )

Line 19. DSM, but the acronym is introduced in line 21

Line 22. Acronym UAV without introduction.

Line 66. LIDAR  ( Lidar Detection And Ranging) it is not correct. LIDAR (LIgth Detection...)

The numeration of the chapters is not correct

  1. Introduction
    • The Vallone dell’Elva Road

  1. Methodology
  2. Results
  3. Discussions
  4. Conclusions

Line 82.  “…both no-contact techniques and traditional support techniques have been applied;” and the abstract :” The paper describes a multi-scale approach carried out by no-contact techniques for...”. Then, the abstract is not exact.

line 117. (upper Triassic) correctly is Upper Triassic

line 121. “the presence of folds and faults”. I think is necessary a brief description or a simplified scheme with a geological structural map.

table 1 Usually appear m3 and m3

Line 163. Digital Surface Models (DSM). But in line 85 the acronym was already introduced.

Line 183. Dips7.0. Correctly Dips 7.0 from Rocscience Engineers

Line 199. “... survey with different techniques TLS (Terrestrial Laser Scanner),” but in line 167 “In this study LIDAR techniques was not applied”. It is confused. If the LIDAR was not applied, it is not necessary to comment it again, it is unnecessary.

Figure 5. The figure includes numbered green symbols, but no reference in the text explains their mean.

Line 226. GNSS (Global Navigation Satellite Station) or Total station. The acronym was already introduced in line 171. Total Station is the correct form.

Line 278. Figures 16 and 17 are mentioned, but the order of the figures is 8. It is not correct, and are mentioned with the incorrect “ at the end of this paper”

Figure 9. The figures shows two outcrops with the denominations A1_b, A3_3, but the figures have not scale, and the position in the area of study are not introduced. The references to each model is introduced forward and the reader lost the references. It is not correct.

Line 401-406. This paragraf of the discussion need a better redaction.

Figure 16 and 17: Text of the figures are illegible

Major

The introduction is too much exhaustive with the description of the in-direct techniques

The parameter roughness is cited in the abstract as a measured parameter

“The paper describes a multi-scale approach carried out by no-contact techniques for the survey of some geometrical features of discontinuities as their orientation, spacing, persistence and roughness useful to identify the possible kinematics and their stability conditions.”  and the introduction: “higher resolution point clouds were obtained to measure the roughness of the discontinuities” but dissapear in methodology and results.

The abstract does not reflect the content of the manuscript. It is focused on the methodology, which is not new, using commercial and well-known software.

An important part of the manuscript is focused on the methodology, which does not present any novelty and could be much more reduced. The discussion and results, which in my opinion is the contribution of the article, is not well explained and little helped with the figures.

The article explains an extensive methodological work to briefly show the results and discussions of the studied area.

Round 3

Reviewer 2 Report

The minor revisions have been corrected, as well as many of the major ones. Certain restructuring and modification of figures, as well as a correct arrangement has corrected many deficiencies in the text. The Conclusions section, in my opinion and as commented in the second review, should include a paragraph of the main purpose of this study (Abstract: analyze how the rock mass geo-structural features can affect the stability slope ...), however all references are on methodology.